# Comparative analysis of automatic segmentation of esophageal cancer using 3D Res-UNet on conventional and 40-keV virtual mono-energetic CT Images: a retrospective study

Hua Zhong[1,*], Anqi Li[1,*], Yingdong Chen[1], Qianwen Huang[1], Xingbiao Chen[2], Jianghe Kang[1] and Youkuang You[3]

[1] Department of Radiology, Zhong Shan Hospital of Xiamen University, School of Medicine, Xiamen University, Xiamen, Fujian, China
[2] Clinical Science, Philips Healthcare, Shanghai, China
[3] Department of Radiology, Xiamen Xianyue Hospital, Xiamen, Fujian, China
[*] These authors contributed equally to this work.

## ABSTRACT

**Objectives**. To assess the performance of 3D Res-UNet for fully automated segmentation of esophageal cancer (EC) and compare the segmentation accuracy between conventional images (CI) and 40-keV virtual mono-energetic images ($VMI_{40\ kev}$).
**Methods**. Patients underwent spectral CT scanning and diagnosed of EC by operation or gastroscope biopsy in our hospital from 2019 to 2020 were analyzed retrospectively. All artery spectral base images were transferred to the dedicated workstation to generate $VMI_{40\ kev}$ and CI. The segmentation model of EC was constructed by 3D Res-UNet neural network in $VMI_{40\ kev}$ and CI, respectively. After optimization training, the Dice similarity coefficient (DSC), overlap (IOU), average symmetrical surface distance (ASSD) and 95% Hausdorff distance (HD_95) of EC at pixel level were tested and calculated in the test set. The paired rank sum test was used to compare the results of $VMI_{40\ kev}$ and CI.
**Results**. A total of 160 patients were included in the analysis and randomly divided into the training dataset (104 patients), validation dataset (26 patients) and test dataset (30 patients). $VMI_{40\ kev}$as input data in the training dataset resulted in higher model performance in the test dataset in comparison with using CI as input data (DSC:0.875 *vs* 0.859, IOU: 0.777 *vs* 0.755, ASSD:0.911 *vs* 0.981, HD_95: 4.41 *vs* 6.23, all $p$-value <0.05).
**Conclusion**. Fully automated segmentation of EC with 3D Res-UNet has high accuracy and clinically feasibility for both CI and $VMI_{40\ kev}$. Compared with CI, $VMI_{40\ kev}$ indicated slightly higher accuracy in this test dataset.

Corresponding authors
Jianghe Kang, 1363644560@qq.com
Youkuang You, yyk577@163.com

## INTRODUCTION

Esophageal cancer (EC) is the seventh most common cancer and sixth leading cause of cancer deaths worldwide in 2020, accounting for one in 18 cancer deaths (*Sung et al., 2021*). At present, surgical treatment remains the main choice for EC, unfortunately EC is frequently found at an advanced stage when surgery alone cannot achieve cure. Despite recent medical advances that have enabled a remarkable increase in overall cancer survival, the prognosis of EC remains poor, with a 5-year survival rate of 20% (*Yan et al., 2019*; *Pennathur et al., 2013*). Although concurrent chemoradiotherapy has been established as a standard treatment option for unresectable EC, radiotherapy is also one of the most important treatment strategies for EC (*Garg et al., 2016*; *Babic, Fuchs & Bruns, 2020*). Thorax CT images are usually used for EC radiotherapy planning. The critical step in the planning pipeline involves the extraction of the esophageal gross tumor volume (GTV) from CT data by manual segmentation. However, manual or even semi-automatic algorithms are usually operator-dependent, time-consuming and subject to high inter and intra-observer variability in clinical practice. Thus, developing an automatic and reliable esophageal GTV segmentation approach is desirable. However, automatic esophageal GTV segmentation of CT scans has been rarely implemented, and is known to be challenging due to various shapes, the poor contrast of the tumor with surrounding tissues, and the existence of foreign bodies in the esophageal lumen.

Currently, the main methods for automatic segmentation of esophageal cancer on CT images include the following: (1) Atlas-based method: by registering the normal atlas to the image to be segmented, the corresponding labels in the atlas are used to segment the lesion. This method is simple and easy to implement, but there are large registration errors and poor segmentation results (*Yang et al., 2017*; *Wang et al., 2015*). (2) Feature-based method: use features such as morphology, gray level, and texture in the image to construct a classifier or cluster for lesion segmentation. Commonly used features include morphological features, texture features, and gray level histogram features. This method is easily affected by image quality and feature selection, and the final effect is difficult to guarantee (*Brady et al., 2021*). (3) Deep learning-based method: use convolutional neural networks for end-to-end learning and segmentation. This method can automatically learn image features for segmentation, and the effect is good, but it requires a large amount of training data. In recent years, with the development of deep learning technology, deep learning algorithms have been widely used in the field of medical image segmentation. U-Net has gradually become a hot spot in the field of image segmentation due to its good segmentation performance (*Ronneberger, Fischer & Brox, 2015*). Several studies have developed specific deep learning-based models relying on a U-Net convolutional architecture for automatic segmentation of the lung, heart and liver, yielding promising results, with a reported Dice Similarity Coefficient (DSC) between 0.92 and 0.95 (*Zhu et al., 2019*; *Ammar, Bouattane & Youssfi, 2021*; *Jin et al., 2020b*). Although there have been some studies on CT-based automatic segmentation of esophageal tumors, automatic segmentation of esophageal tumors in CT images is still a challenging task due to the low contrast between the tumor and surrounding tissues. This field is still in the exploratory stage and requires further

research to improve segmentation accuracy and better apply it in clinical practice (Yousefi et al., 2021; Jin et al., 2022; Yue et al., 2022).

Dual-layer spectral-detector computed tomography (SDCT) can acquire temporal and spatial matched high- and low-energy photons from the same X-ray source, which also enables material decomposition in the projection domain. The anti-correlated noise suppression and iterative reconstruction algorithms are also applied, resulting in image noise reduction and image quality improvement (Ozguner et al., 2018; Lu et al., 2019; Kim et al., 2019). Virtual monoenergetic imaging (VMI) is a technique used in computed tomography (CT) imaging to generate images at a specific energy level, also known as a monoenergetic image. VMI images are derived from a spectral CT acquisition that captures the attenuation of X-ray photons at different energy levels. VMI are equivalent to single-energy radiographs, including 161 energy levels ranging from 40 to 200 keV. Certainly, the VMI 40 keV has a good soft tissue resolution and maintains low noise, which makes it suitable for routine lesion detection and observation of subtle differences in soft tissue. Bruns et al. (2020) showed that virtual mono-energetic images improve the accuracy of a fully convolutional network used for myocardial segmentation on cardiac CT angiography (CCTA) images, with the DSC increased from 0.846 to 0.901 compared with only conventional CCTA images. Dima et al. (2021) trained a U-Net based model to segment peripancreatic arteries on iodine material decomposition images, and the final DSC exceeded 0.95. These studies implied that spectral images obtained using SDCT can provide additional information to optimize the algorithms. To our knowledge, it has not been investigated whether the automatic segmentation model performance can be improved using $VMI_{40\ keV}$ compared to CI.

The main purpose of this study was to evaluate the performance of the 3D Res-UNet for fully automated segmentation of EC on SDCT scans acquired for radiotherapy treatment planning, and compare performance between CI and $VMI_{40\ keV}$ data.

## MATERIALS AND METHODS

All procedures performed in studies involving human participants were in accordance with the ethical standards of the institutional. Informed consent was waived by Institutional Review Board due to retrospective study characteristics.

### Patient enrollment

This retrospective study was approved by the institutional review board of Zhongshan Hospital affiliated to Xiamen University, and informed consent was waived. The Ethical Approval number was XMZSYY-AF-SC-12-03. 259 Patients with histologically confirmed EC by biopsy or surgery, who underwent enhanced CT with SDCT from May 2019 to December 2020 were selected for this study. Exclusion criteria were: (a) chemotherapy or radiotherapy or other anticancer treatments before the IQon CT scans; (b) a history of other malignancies; and (c) incomplete medical records. After excluding 99 cases, 160 patients were finally enrolled in this study and randomized divided into the training dataset (104 patients), validation dataset (26 patients) and test dataset (30 patients). Figure 1 depicts the process used to select the images.

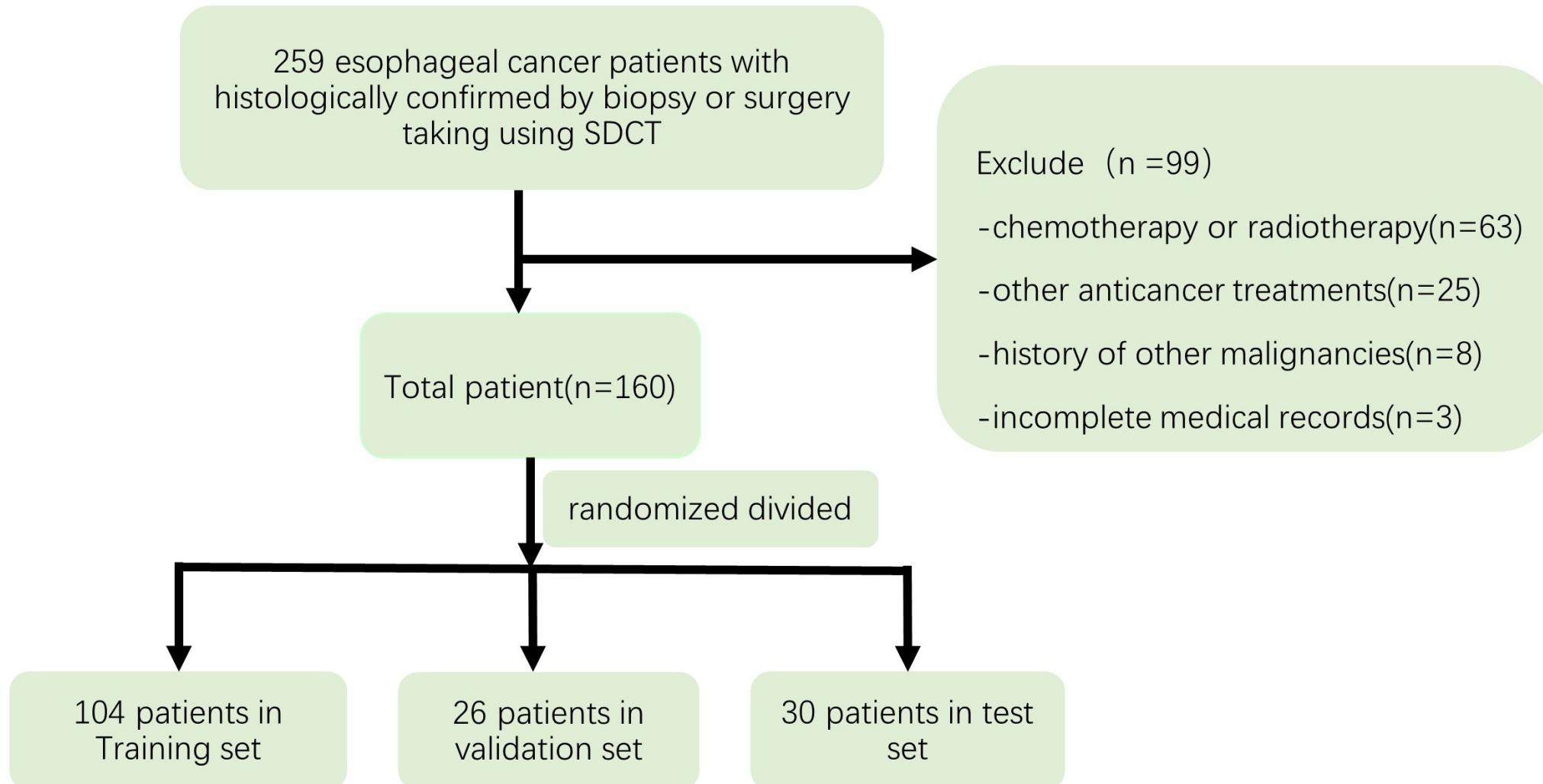

**Figure 1** **Flow chart depicting the selection of the study patients.**

## CT image acquisition

Chest CT was performed in both the arterial and venous phases on a SDCT (IQon Spectral CT, Philips Healthcare, Best, The Netherlands). All the patients were in the supine position, with both arms raised and breathing calmly; while holding the breath, scanning range was from the epiglottis to the costophrenic angle to ensure coverage of all esophageal and lung tissues. Before initiating contrast-enhanced scanning, 70 ml contrast medium (Ultravist injection at a concentration of 300 mg/ml; Bayer, Leverkusen, Germany) was injected through the elbow vein at a rate of 2.5 ml/s using a power injector. After the injection, 20–30 ml of normal saline was injected for flushing at a rate of 3 ml/s. Arterial and venous phase images were acquired at 25s and 60s. The following scanning parameters were used: detector collimation, 64 × 0.625 mm; tube voltage, 120 kV; tube current, 180–250 mA; rotation speed, 0.33s/rot; helical pitch, 0.671; matrix, 512 × 512.

## Image reconstruction

Spectral base image (SBI) datasets were reconstructed, with a spectral level of three, a slice thickness of one mm, a slice increment of one mm, standard kernel (B) with a window width of 400 HU & a level of 40 HU, and sharp kernel (YB) with a window width of −1400 HU & a level of −600 HU.

All arterial phase images were transferred to the dedicated workstation (IntelliSpace V9, Philips Healthcare) for further analysis. Both conventional images (CI) and virtual mono-energetic images at 40 keV ($VMI_{40\ keV}$) were derived from SBI.

## Esophageal cancer annotation

An open-source software (3D slicer, version 4.11, https://www.slicer.org/) was used to delineate EC contours manually. All the tumor contours were drawn on each slice of $VMI_{40\ kev}$ by a radiologist with 8 years of experience and peer-reviewed by a senior

radiologist with 15 years of experience. The adjacent airway, lymph nodes, heart, vascular structures and thoracic vertebrae were avoided during contour drawing. Dice similarity coefficient (DSC) was performed to assess consistency between the two observers for EC segmentation.

## Image pre-processing

Image pre-processing was performed as follows. (1) All voxel sizes were reset to 1 mm × 1 mm × 1 mm by linear interpolation, and matrices were defined as n ×150×180 (slices ×columns ×rows) by Simple ITK (version 2.1, https://simpleitk.org/) just for the inclusion of the EC and a small amount of adjacent tissues, to reduce the calculation burden and graphic card's memory storage. (2) The ScaleIntensityRanged function in MONAI (version 0.6, https://monai.io/) was used to rescale the density values of our images and all density values in our images which ranged from −50 HU (minimum) to 400 HU (maximum), were normalized and artificially scaled to a range of 0 to 1 using this function. (3) Before sending the images into a convolutional neural network, all images in the training dataset were randomly cropped to size of 96 × 96 × 96 $mm^3$ and data augmentation was achieved in the training dataset by image flipping and rotation using the RandCropByPosNegLabeld and RandAffined functions in MONAI.

## Automatic segmentation model and training

In this study, 3D Res-UNet, the classical U-net scheme combined with a residual connection unit (RCU), was used for the automatic segmentation task. Five residual units were added to the automatic segmentation model in this study, The model uses a kernel size of (3, 3, 3) and varying number of channels (16, 32, 64, 128, 256) to build the layers of the encoder and decoder. It uses a stride of two to downsample the spatial dimensions of the encoder and residual units with batch normalization to improve the model's robustness. The architecture of the automatic segmentation model is shown in Fig. 2

The Adam optimizer was used to train the automatic segmentation model with the Dice loss function as the cost function. The batch size was set to eight, the learning rate was set to 0.0001. The model was trained for 600 epochs. The curves depicting the training process of the networks are presented in both Fig. 3.

The model was built using PyTorch (version 1.8, https://pytorch.org/) and MONAI (version 0.6, https://monai.io/) and trained on a Linux workstation (Ubuntu version 20.04) with one NVIDIA GeForce GTX3090 GPU with 24 GB memory (NVIDIA, Santa Clara, VA, USA).

## Evaluation of segmentation performance in the test dataset

After training the model, we examined its performance in predicting the ROIs in the test dataset. Segmentation accuracy was evaluated using the DSC as follows. (1) A statistical measure of spatial overlap also defined as DSC was derived as 2TP/(FP+2TP+FN), where TP, FP and FN are the numbers of true positive, false positive and false negative detections, respectively. (2) Intersection of union (IOU) was defined as TP/(FP+TP+FN), which evaluates the numbers of TP and FN detections. (3) Average Symmetric Surface Distance (ASSD) was defined as $\text{ASSD(A,B)} = \frac{1}{S(A)+S(B)}\left(\sum_{sA\in S(A)} d(sA, S(B)) + \sum_{s_B\in S(B)} d(sB, S(A))\right)$, where S(A) denotes the set of surface voxels of A. The shortest distance of an arbitrary

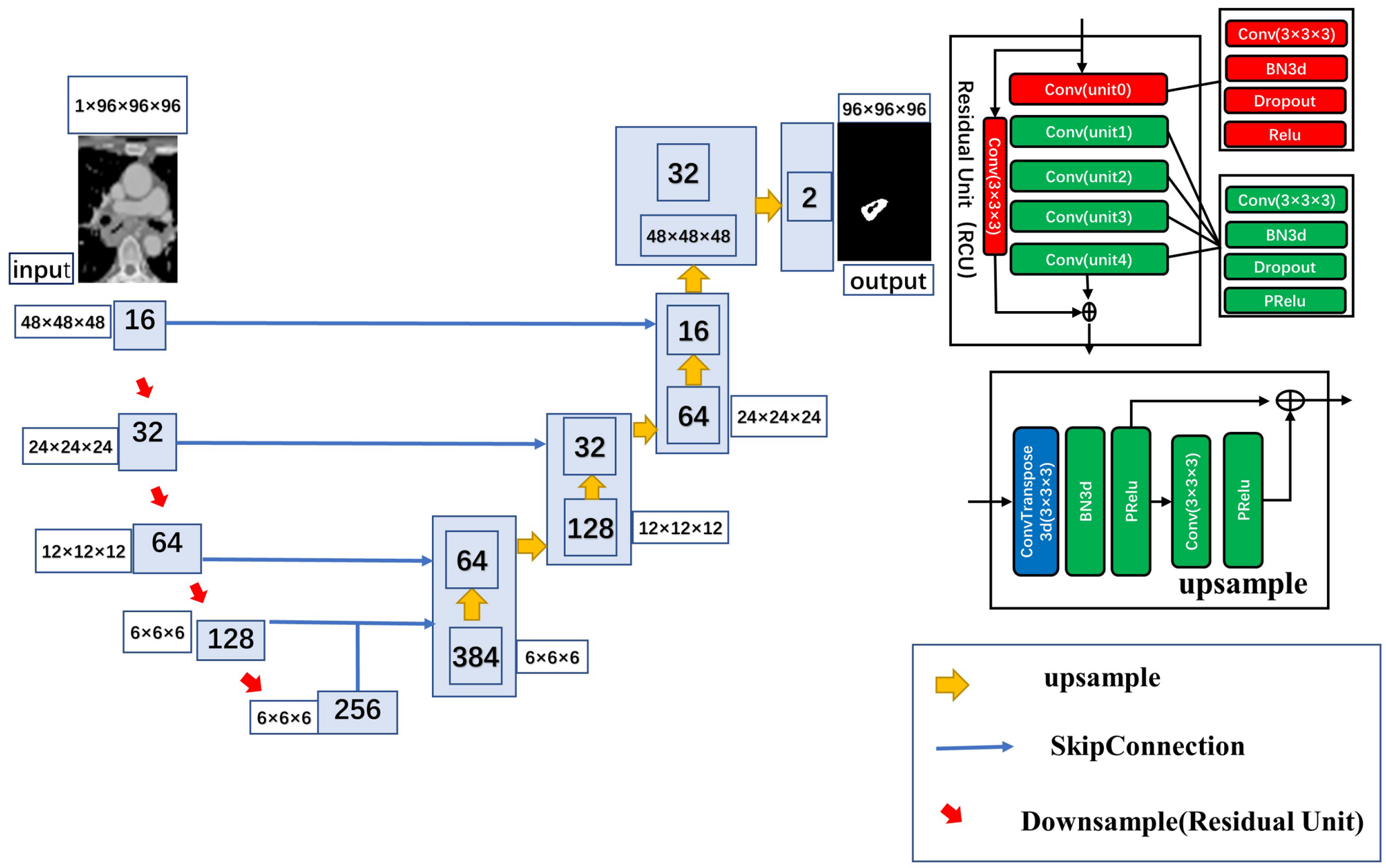

Figure 2 Detailed architecture of the 3D ResUnet.

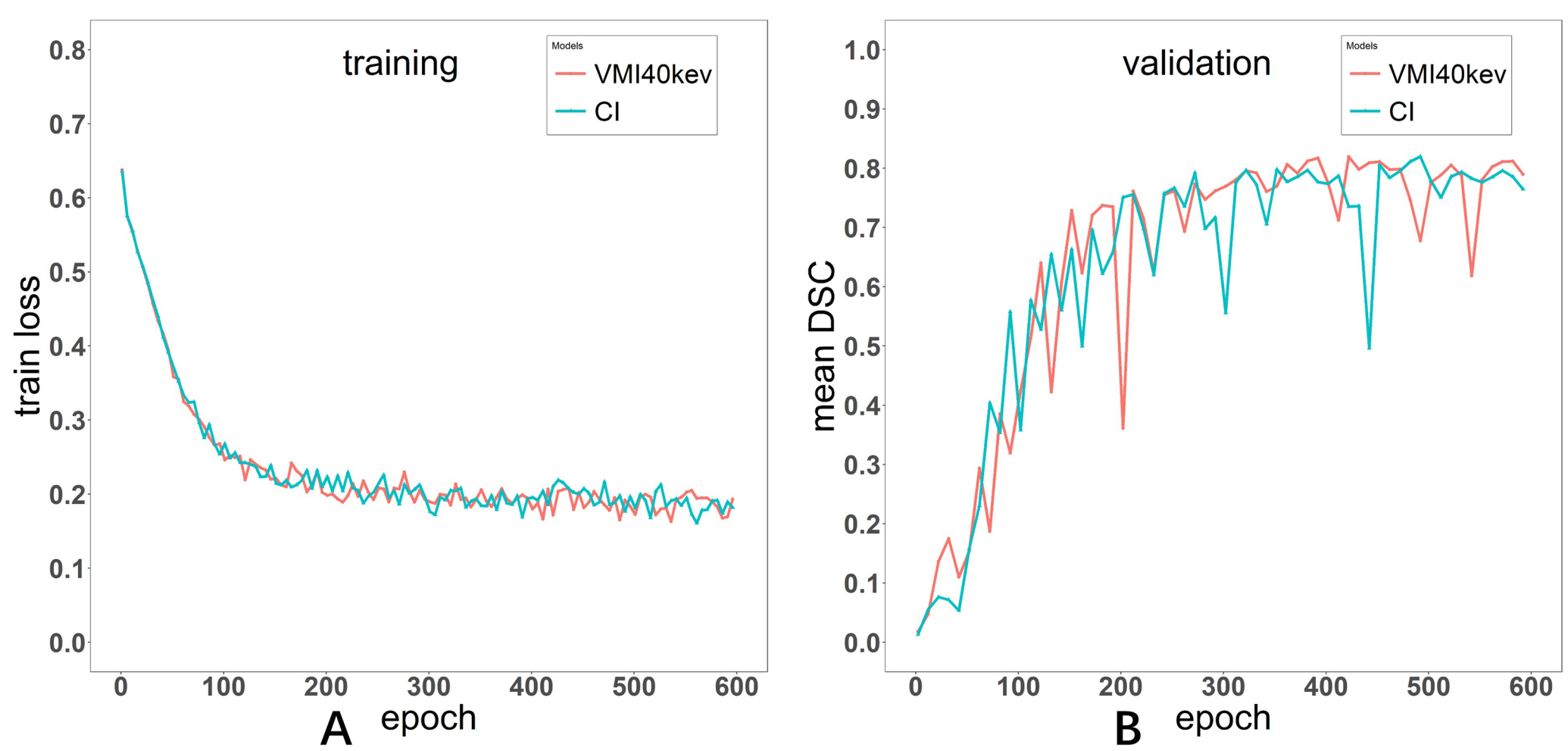

Figure 3 Loss curve of the training process for Models $VMI_{40\ keV}$ and CI (A). Evaluation metric curve of the training process for $VMI_{40\ keV}$ and CI (B).

voxel $v$ to S(A) was defined as: $d(v, S(A)) = \min_{s_A \in S(A)} \|v - s_A\|$, where ‖.‖ denotes the Euclidean distance. (4) 95% Hausdorff Distance (HD_95) was defined as H(A,B) = max(h(A,B), h(B,A)), a measure that describes the 95th percentile of all distances between points in image A and the nearest point in image B, where h(A,B) = max(a ∈A)min(b ∈B) ‖a-b ‖ and h(B,A) = max(b ∈B)min(a ∈A) ‖b-a ‖ and ‖b-a ‖ is the Euclidean distance.

**Table 1 Participant demographics.**

| Variable | Train and validation | Test | *P* value |
|---|---|---|---|
| n | 104 + 26 | 30 | |
| Age (year) | 63.7 ±9.1 | 63.7 ±10.1 | 0.999 |
| Sex Male | 98 | 26 | 0.275 |
| Female | 32 | 4 | |
| Histopathology | | | |
| Squamous cell | 126 | 28 | |
| Non-Squamous cell | 4 | 2 | 0.689 |
| Grade of differentiation | | | |
| Well/moderate | 115 | 28 | |
| poorly | 15 | 2 | 0.651 |
| Location | | | |
| Cervical upper and upper+ middle | 79 | 19 | |
| Middle +lower and lower | 51 | 11 | 0.795 |
| Tumor size($mm^3$) | 24.97(12.56,34.99) | 21.34(14.23,40.48) | 0.481 |
| T stage | | | |
| 2 | 10 | 3 | |
| 3 | 97 | 24 | |
| 4 | 23 | 3 | 0.565 |

**Notes.**
Categorical factors are displayed as n and continuous factors are displayed as mean ±SD or interquartile ranges.

## Statistical analysis

Statistical analyses were performed with R (version 4.0; *R Core Team, 2020*). Patients' age in the training plus validation datasets *versus* test dataset were compared with two-tailed Independent *t*-test. The sex, histopathological features, differentiation grade and T stage of EC were compared by the two-sided Fisher's exact test. The independent sample rank-sum test was used for comparing tumor size. DSC, IOU, ASSD and HD_95 in the test dataset with $VMI_{40\ keV}$ and CI as input data were compared by the paired Wilcoxon signed-rank test. $P < 0.05$ was considered statistically significant.

# RESULTS

## Patient characteristics

A total of 160 patients were included in the analysis. They were aged from 39 to 88 years (mean of 64 years). Table 1 lists clinical and demographic features in the training and validation and test datasets. The results showed no statistically significant differences in clinical and demographic characteristics between the training and validation and test datasets. The histopathological types were squamous cell carcinoma, adenocarcinoma and small cell carcinoma, most of which were well differentiated or moderately differentiated. Tumor sizes ranged from 4.16 $cm^3$ to 142.82 $cm^3$ (median of 24.97 $cm^3$). The DSC for the EC segmentation between the two observers was $0.898 \pm 0.036$.

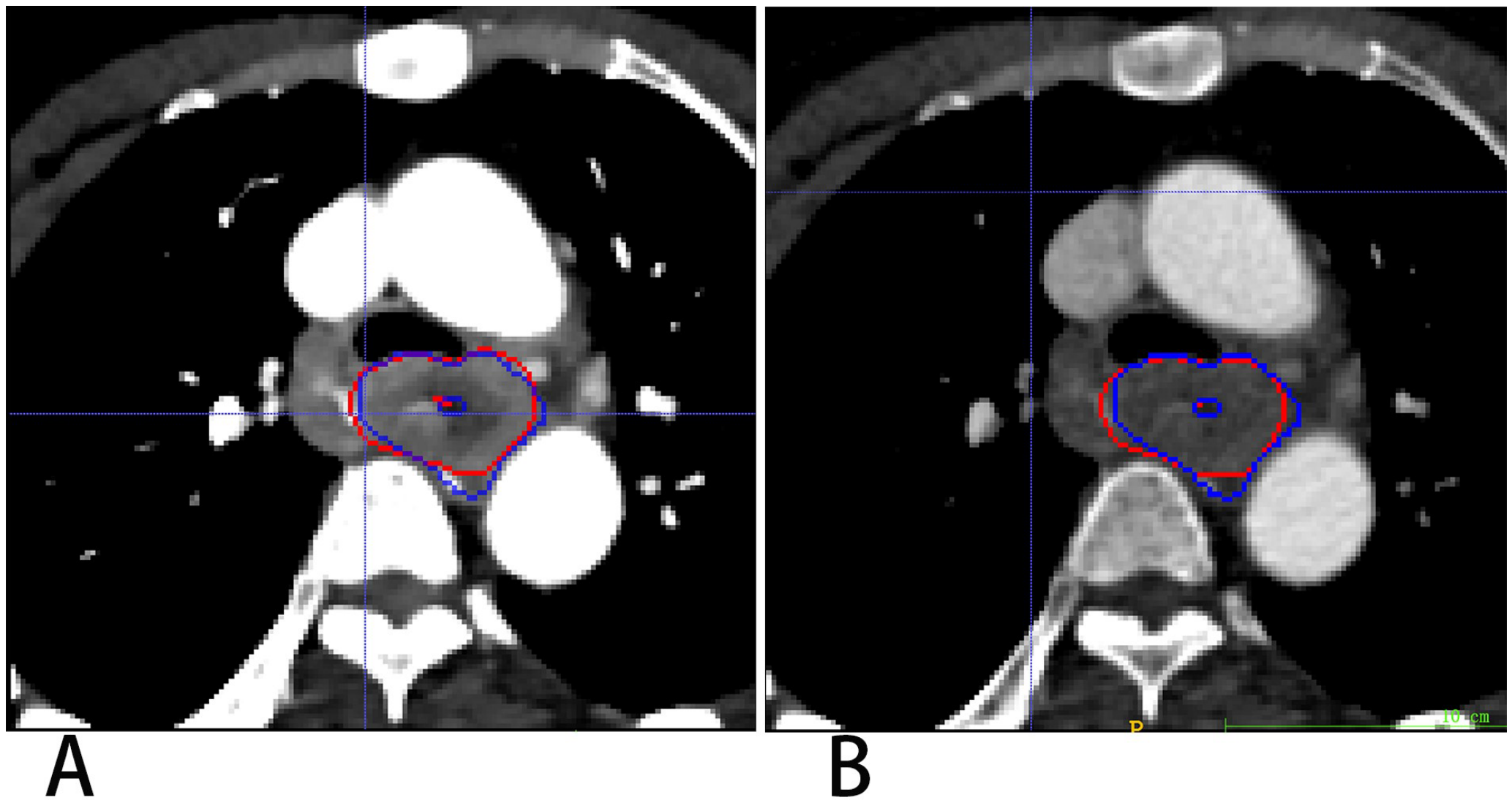

**Figure 4 Segmentation results on $VMI_{40\ keV}$ and CI $VMI_{40\ keV}$ (A) and CI (B).** Red and blue contours are label and segmentation, respectively.

**Table 2 DSC, IOU, ASSD and HD_95 of EC for $VMI_{40\ keV}$ and CI in the test dataset using the automatic segmentation model.**

| Metric, median (Q1, Q3) | $VMI_{40\ keV}$ | CI | *P*-value |
|---|---|---|---|
| DSC | 0.874(0.835,0.901) | 0.859(0.787, 0.899) | 0.022 |
| IOU | 0.777(0.716,0.820) | 0.755(0.651,0.817) | 0.021 |
| ASSD (mm) | 0.911(0.619, 1.296) | 0.981(0.627,1.762) | 0.028 |
| HD_95 (mm) | 4.41(3.00, 9.25) | 6.23(3.45, 14.94) | 0.035 |

## Model performance

3D Res-UNet was applied to automatically segment the target volumes of EC on CT images. Segmentation results for a representative patient for $VMI_{40\ keV}$ and CI are shown in Fig. 4. The median results of DSC, IOU, ASSD and HD_95 statistics for the segmentation indexes $VMI_{40\ keV}$ and CI in the test dataset are shown in Table 2, while Fig. 5 shows the distribution of the results. We trained and evaluated 3D Res-UNet on $VMI_{40\ keV}$ and CI, separately. There were statistically significant differences in DSC, IOU, ASSD and HD_95 between the models with $VMI_{40\ keV}$ and CI. The results of $VMI_{40\ keV}$ were slightly better for the CI with the automatic segmentation model in DSC, IOU, ASSD and HD_95. According to the good agreement threshold of the DSC (DSC $\geq$0.7) (*Zijdenbos et al., 1994*; *Bartko, 1991*), automatic segmentation results for the EC of $VMI_{40\ keV}$ and CI were consistent with manual segmentation results.

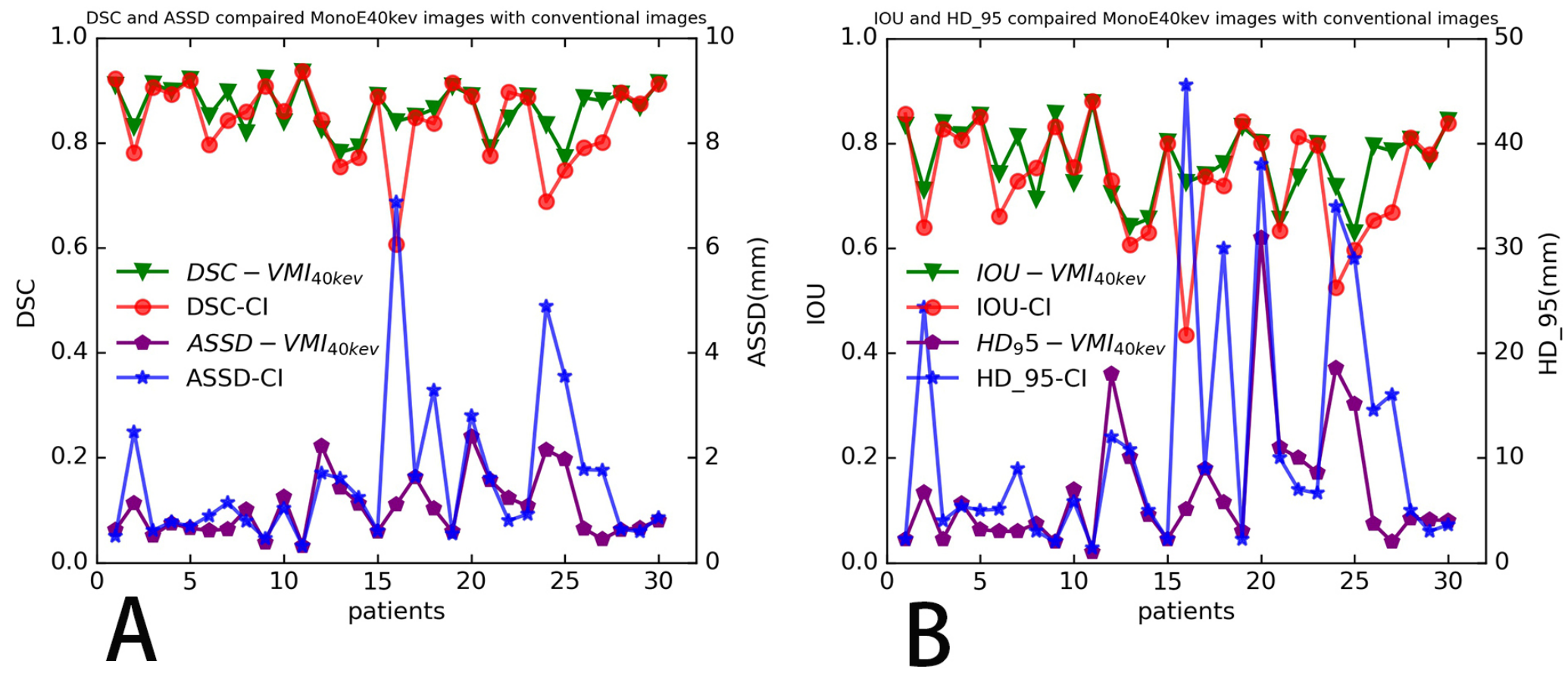

**Figure 5 Four indicators to evaluate the segmentation performance of automatic models distribution of four evaluation metrics.** (A) DSC and ASSD, (B) IOU and HD_95 of ROIs determined in test dataset (30 patients), for each segmentation mask (no patient's dice below 0.5).

## DISCUSSION

In this study, the 3D Res-UNet convolutional neural network (CNN) algorithm was trained for EC segmentation using enhanced $VMI_{40\ keV}$. After training, the CNN model could be used for the automatic segmentation of EC on both enhanced $VMI_{40\ keV}$ and CI with a median DSC greater than 85%.

CNN architectures are supervised models that are trained end to end to learn a hierarchy of features representing different levels of abstraction in a data-driven manner; therefore, good data determine a good model. There is a lack of soft tissue contrast on CT images, some of which are unsatisfactory. *Yousefi et al. (2018)* presented a 3D end-to-end method based on a CNN for esophageal gross tumor volume segmentation, which achieved a DSC value of 0.73 ± 0.20. *Jin et al. (2020a)* introduced a progressive semantically-nested network segmentation model for gross tumor volume segmentation of esophageal cancer with a DSC value of 0.751. *Huang et al. (2020)* investigated a Channel-Attention U-net for semantic segmentation of the esophagus and EC, with a value for IOU in EC of only 0.625. Table 2 shows our model had a median IOU of 0.777 for $VMI_{40\ keV}$ and 0.755 for CI, indicating its superiority over the previously reported models. Our model achieved a median DSC of 0.874 for $VMI_{40\ keV}$ and 0.859 for CI, further improving the accuracy of automatic segmentation, which may be explained by that better data with best contrast to noise ratios (CNRs) were obtained in this study to train a better model drawing the tumor contours on $VMI_{40\ keV}$ arterial phase images (*Kang et al., 2019*; *Mochizuki et al., 2022*).

Although SDCT can provide both VMI and CI, the patients were scanned by conventional CT during daily work. In this research, we trained the CNN with same label on both $VMI_{40\ keV}$ and CI produced by SDCT, and compared their segmentation accuracies, achieving median DSCs of 0.874 for $VMI_{40\ keV}$ and 0.859 for CI. This means the model trained on CI also had a good segmentation performance, supporting the use of the trained

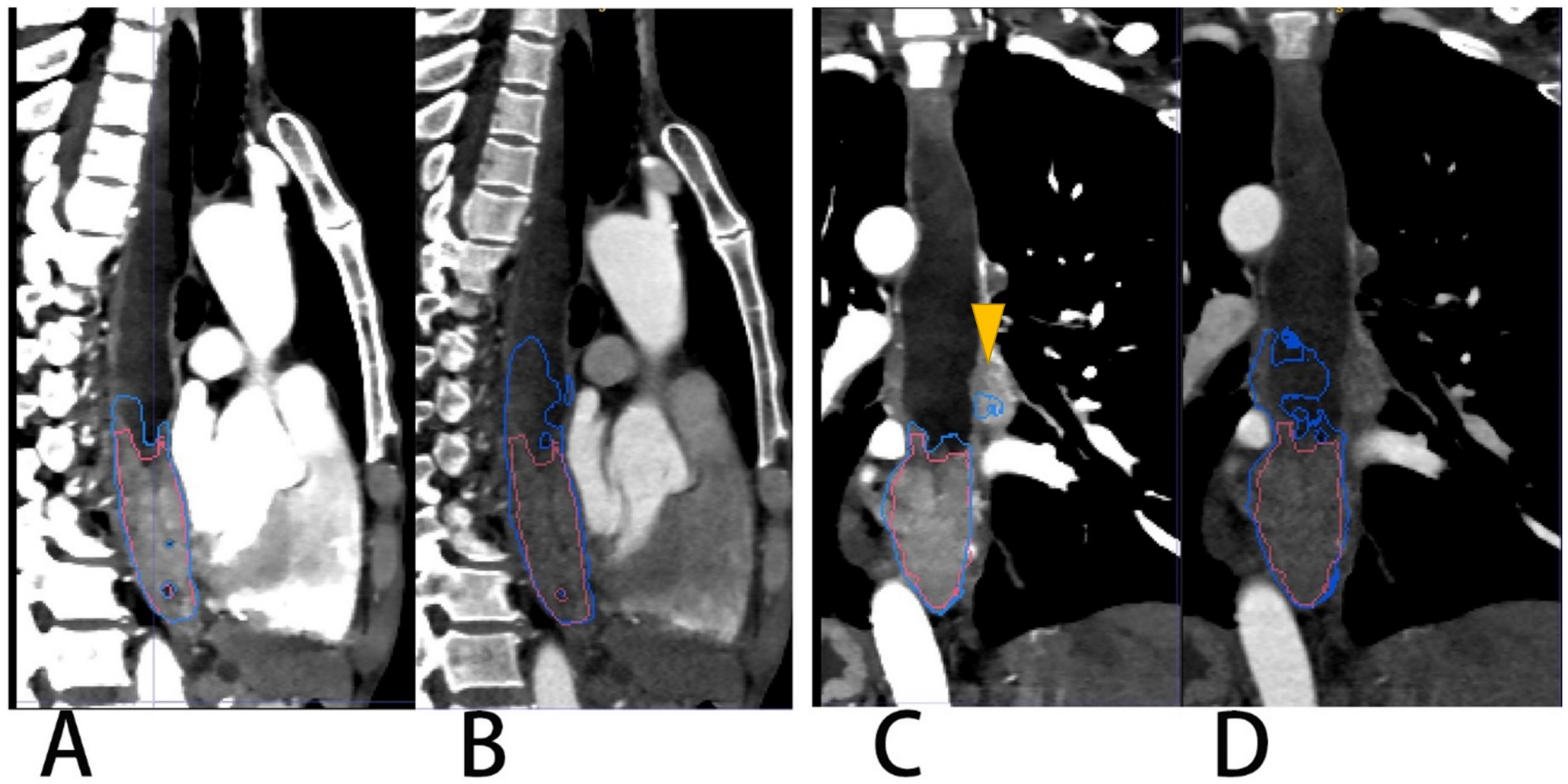

**Figure 6 A patient with EC located in the lower thoracic segment leading to obstruction and effusion of the upper esophagus.** (A & C) VMI40kev. (B & D) CI. Red and blue contours are label and segmentation, respectively. The yellow triangular arrow points to lymph nodes in (C). The segmentation on CI of (B) and (D) shows an overestimation of the EC.

CNN for automatic segmentation of organs or tumors on outside-sourced conventional CT images.

According to this research, the difference in the segmentation performance of DSC was statistically significant ($p < 0.05$) between $VMI_{40\ keV}$ and CI (median DSC 0.874 *vs.* 0.859), and similarly as IOU, ASSD and HD_95. The possible reasons are that low-keV VMI significantly improved attenuation and signal-to-noise ratio for the primary tumor, and the CNR of the primary tumor *vs.* circumjacent anatomical structures (esophageal wall, diaphragm and liver parenchyma) was significantly higher in low-keV VMI, peaking at $VMI_{40\ keV}$ (*Lee et al., 2018*). Figure 6 shows the segmentation contour of EC between $VMI_{40\ keV}$ and CI, as $VMI_{40\ kev}$ highlighted the edges of the soft tissue, especially when the obstruction of esophageal cancer leads to the dilatation and effusion of the upper esophagus; thus, the DSC of EC was improved by 0.148. However, this may cause another problem: other tissues such as lymph nodes adjacent to the EC may be regarded as part of the EC. Using $VMI_{40\ keV}$ as input data improved the accuracy of segmentation.

The overall effect of the experiment was good both for $VMI_{40\ keV}$ and CI. According to Table 2, automatic segmentation had a more stable performance for $VMI_{40\ keV}$ compared with CI. There was a segmentation effect of our network on some small boundaries when the EC had a cavity, especially on conventional images. Figure 7 shows a patient in whom the upper half of the EC like the wall around was discarded. Our network achieved the worst segmentation performance with a DSC of 0.606 for CI, which was improved to 0.841 for $VMI_{40\ keV}$. Unfortunately, the part of the EC adjacent to the thoracic vertebrae was too thin, and was abandoned both on $VMI_{40\ keV}$ and CI.

Besides its retrospective design, this study had some limitations. First, it had a single-center design and used a single CT scanner. As only data from our institution were applied,

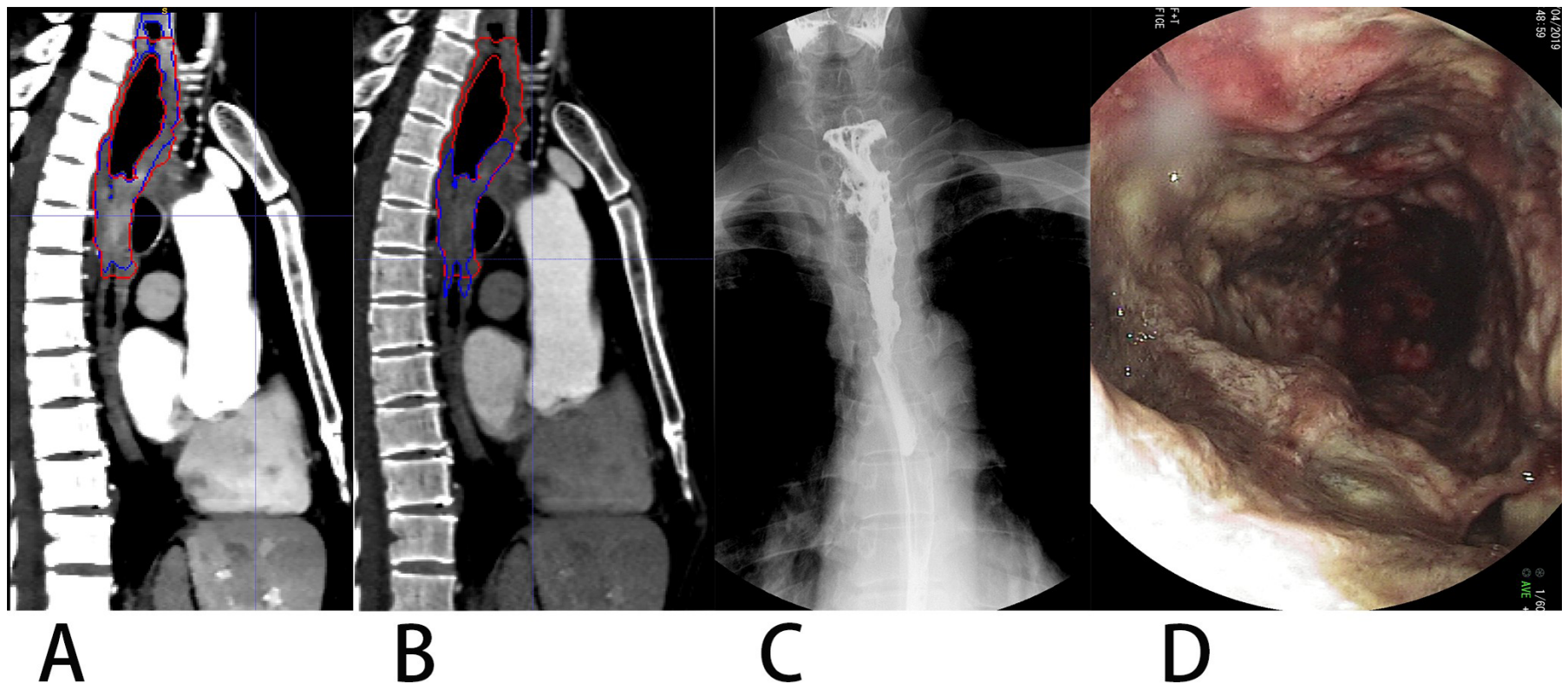

**Figure 7 A patient with upper thoracic EC showing constricted portion (lower) and non-constricted portion (upper).** (A) VMI40kev, (B) CI. Spot film of the barium-swallow study (C) and endoscopic image (D) indicated the upper affected but non-constricted portion of EC. Image B shows missed segmentation of the non-constricted upper portion on CI.

generalization to other scanners and sites requires the testing of the generalizability of our automatic segmentation model. Secondly, the sample size of this study was limited. A network is affected by the sample size, and the generalization ability of the model has not been fully verified. Some cases with poor segmentation were found in EC; because of the limited cases included in the current study, more cases are needed to improve the accuracy of automatic segmentation in these cases. In the future, we plan to apply our model to datasets obtained with various CT scanners in various institutions. Thirdly, metastatic lymph node was also required to delineate along with EC lesion for radiotherapy treatment planning. While our primary study only focused on the EC lesion segmentation.

## CONCLUSION

In conclusion, automatic EC segmentation is clinically feasible with 3D-ResUNet, and using $VMI_{40\ keV}$ can increase its accuracy. Fully automated localization and segmentation of EC based on the present study could be valuable for detection, diagnosis and chemoradiation therapy planning in the future.

### Funding

The authors received no funding for this work.

### Competing Interests

Xingbiao Chen is an employee of Philips Healthcare, the manufacturer of the scanner.

## Author Contributions

- Hua Zhong conceived and designed the experiments, performed the experiments, analyzed the data, prepared figures and/or tables, authored or reviewed drafts of the article, and approved the final draft.
- Anqi Li conceived and designed the experiments, performed the experiments, analyzed the data, prepared figures and/or tables, authored or reviewed drafts of the article, and approved the final draft.
- Yingdong Chen performed the experiments, prepared figures and/or tables, and approved the final draft.
- Qianwen Huang performed the experiments, authored or reviewed drafts of the article, and approved the final draft.
- Xingbiao Chen performed the experiments, prepared figures and/or tables, and approved the final draft.
- Jianghe Kang conceived and designed the experiments, prepared figures and/or tables, authored or reviewed drafts of the article, and approved the final draft.
- Youkuang You performed the experiments, prepared figures and/or tables, authored or reviewed drafts of the article, and approved the final draft.

## Ethics

The following information was supplied relating to ethical approvals (i.e., approving body and any reference numbers):

The Institutional Review Board of Zhongshan hospital affiliated to Xiamen University approved the study (XMZSYY-AF-SC-12-03).

## Data Availability

The code and raw data are available in the Supplemental Files.

## Supplemental Information

Supplemental information for this article can be found online at http://dx.doi.org/10.7717/peerj.15707#supplemental-information.

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
