# Peer review of "Comparative analysis of automatic segmentation of esophageal cancer using 3D Res-UNet on conventional and 40-keV virtual mono-energetic CT Images: a retrospective study"

_PeerJ, doi:10.7717/peerj.15707_

## Round 0.1 · original submission · Major Revisions

· Academic Editor

Major Revisions

Please carefully consider the comments raised by reviewers.

Reviewer 1 ·

Basic reporting

The structure of this article is well organised. Esophageal cancer (EC) is the seventh most common cancer. Radiotherapy is one of the most important treatment strategies for EC. Thorax CT images are usually used for EC radiotherapy planning. The critical step in the planning pipeline involves the extraction of the esophageal gross tumor volume (GTV) from CT data by manual segmentation. However, manual or even semi-automatic algorithms are usually operator-dependent, time-consuming and subject to high inter and intra-observer variability in clinical practice. Besides, the spectral images obtained using SDCT can provide additional information to tranditional CT. Therefore, the authors evaluated the performance of deep learning model for fully automated segmentation of EC on SDCT scans acquired for Radiotherapy treatment planning, and compare performance between CI and VMI40kev data. They found that the VMI 40kev as input data resulted in higher learning efficacy in the training datasets and slightly higher accuracy in the test datasets in comparison with CI.

Experimental design

1. What are the criteria for dividing the training, validation and test sets?
2. "VMI 40kev as input data resulted in higher learning efficacy in the training datasets", how to define the higher learning efficacy.
3. "Fully automated segmentation of EC with 3D Res-UNet has high reliability", how reliability is demonstrated
4. "However, very few automatic segmentation models for EC have been proposed based on CT images", a brief summary of the results of these few relevant studies is suggested.
5. Can the inclusion and exclusion criteria for the sample be represented in a flow chart?
6. "All density scopes were -50 HU (min) to 400 HU (max); the density was normalized and artificially set from 0 to 1.", can you tell me exactly what normalization method was used and if you can give me the formula?
7. Can the image pre-processing code be provided, or at least the software, toolkit and version in use?
8. It is recommended that loss convergence plots are provided for the training and validation sets.
9. "...in the train and validation datasets and test datasets were compared by the t-test.", does the t-test here refer to a two-sample t-test? Is it a two-tailed.
10. Is there a test for consistency between the two observers when EC segmentation?
11. What does each number in Table 1 mean when statistically describing the size of the tumour?

Validity of the findings

The image pre-processing code should be provided, or at least the software, toolkit and version in use?

Reviewer 2 ·

Basic reporting

In this study, the authors use 3D Res-UNet to assess the performance of fully automated segmentation of esophageal cancer. The experimental results show the 3D Res-UNet works well on CI and VMI_40kev data. It is also proved that the VMI_40kev is helpful for modelling the segmentation performance. In addition, the authors demonstrated that the 3D Res-UNet is clinically feasible and has practical value in planning patient detection, diagnosis and radiotherapy.
The paper is well organized into sections and the presentation is clear to me. I think the paper is suitable for publication in PeerJ if the authors can address the comments properly.

Experimental design

(1) Please describe the 3D Res-UNet model in detail, including the convolution strides, paddings, and batch sizes, etc.
(2) Although this paper compares the results from other papers, it does not use its own data to compare the results of the classical or state-of-the-art models with 3D Res-UNet, which fails to comprehensively assess the superiority of the 3D Res-UNet in the segmentation of esophageal cancer.
(3) Why is the epoch set to 600? In my opinion, such an epoch can lead to a big computational cost. Why not change the learning rate during training phase? Can the learning rate of 0.0001 reach the optimal value? I suggest the authors consider the dynamic learning rate.
(4) How is the deconvolution layer in the decoder defined. Will the directly deconvolving the connected tensor cause the deficiency of data fusion, thus resulting in missing feature vectors?

Validity of the findings

(1) Why do the authors give the median of each evaluation index instead of the average? As shown in Figure 3, the split performance for the test set is not very stable. The authors should explain the reason for this?
(2) In the paper, the authors mentioned that low-kev VMI can significantly improve the attenuation and signal-to-noise ratio for the primary tumor. Also, the CNR of the primary tumor vs circumjacent anatomic is significantly higher in the low-kev VMI, peaking at VMI_40kev. It can be seen that VMI_40kev should have a better segmentation than CI. So what is the significance of this study?

Additional comments

no comment

Reviewer 3 ·

Basic reporting

Clear and unambiguous, professional English used throughout.

Experimental design

Research question well defined, relevant & meaningful.

Validity of the findings

Conclusions are well stated, linked to original research question & limited to supporting results.

Additional comments

This study mainly utilized the 3D Res-UNet technique to compare the performance of conventional images and 40-keV virtual monoenergetic images (VMI) in automatic segmentation of esophageal cancer (EC). It was found that using 40-keV VMI can improve the accuracy of EC segmentation. Meanwhile, the 3D Res-UNet method showed better performance in EC detection compared to traditional imaging methods. These findings may contribute to the improvement of diagnostic and therapeutic approaches for EC. However, there are still some minor issues in this study that need further refinement.
Introduction:
1.Please list the methods used in previous studies to segment esophageal lesions and their respective advantages and disadvantages. After analyzing these previous studies, explain why this method was chosen for the present study, its advantages, and the problems it can solve.
2.Please add the relevant introduction for VMI40kev.
3.Please confirm again whether 'informed consent was waived' is correct.
4."Automatic Segmentation Model and training" section could provide more detailed data processing steps.
5.Please thoroughly check for minor grammatical errors throughout the entire manuscript.

---

## Round 0.2 · Minor Revisions

· Academic Editor

Minor Revisions

Some minor concerns should be addressed.

Reviewer 1 ·

Basic reporting

The authors have replied and addressed all the comments I raised. Thank the authors for their work. I recommend publishing this article.

Experimental design

Has been improved.

Validity of the findings

Has been improved.

Reviewer 2 ·

Basic reporting

The authors have addressed all my comments carefully by supplementing experiments and explanation of their model. Also, the presentation is improved significantly in the revised manuscript. In my opinion, the paper is suitable for publication in PeerJ.

Experimental design

no comment

Validity of the findings

no comment

Reviewer 3 ·

Basic reporting

Author has made the modifications as per the suggested revisions, but couldn't find the keywords in the text. Please add the keywords.

Experimental design

Author has made the modifications as per the suggested revisions, but couldn't find the keywords in the text. Please add the keywords.

Validity of the findings

Author has made the modifications as per the suggested revisions, but couldn't find the keywords in the text. Please add the keywords.

---

## Round 0.3 · accepted · Accept

· Academic Editor

Accept

The manuscript can be accepted now.